# NLRP3 Inflammasome and Pyroptosis in Liver Pathophysiology: The Emerging Relevance of Nrf2 Inducers

**DOI:** 10.3390/antiox11050870

**Published:** 2022-04-28

**Authors:** Laura Hurtado-Navarro, Diego Angosto-Bazarra, Pablo Pelegrín, Alberto Baroja-Mazo, Santiago Cuevas

**Affiliations:** 1Molecular Inflammation Group, Biomedical Research Institute of Murcia (IMIB), University Clinical Hospital Virgen de la Arrixaca, 30120 Murcia, Spain; laura.hurtado1@um.es (L.H.-N.); dangosto@um.es (D.A.-B.); pablo.pelegrin@imib.es (P.P.); 2Department of Biochemistry and Molecular Biology B and Immunology, Faculty of Medicine, University of Murcia, 30100 Murcia, Spain

**Keywords:** liver diseases, fibrosis, NLRP3 inflammasome, pyoptosis, gasdermin, Nrf2, ROS

## Abstract

Inflammasomes, particularly the nucleotide-binding oligomerization domain, leucine-rich repeat, and pyrin domain containing 3 (NLRP3) inflammasome, apparently serve as crucial regulators of the inflammatory response through the activation of Caspase-1 and induction of pro-inflammatory cytokines and pyroptotic cell death. Pyroptosis is a type of programmed cell death mediated by Caspase-1 cleavage of Gasdermin D and the insertion of its N-terminal fragment into the plasma membrane, where it forms pores, enabling the release of different pro-inflammatory mediators. Pyroptosis is considered not only a pro-inflammatory pathway involved in liver pathophysiology but also an important pro-fibrotic mediator. Diverse molecular mechanisms linking oxidative stress, inflammasome activation, pyroptosis, and the progression of liver pathologies have been documented. Numerous studies have indicated the protective effects of several antioxidants, with the ability to induce nuclear factor erythroid 2-related factor 2 (Nrf2) activity on liver inflammation and fibrosis. In this review, we have summarised recent studies addressing the role of the NLRP3 inflammasome and pyroptosis in the pathogenesis of various hepatic diseases, highlighting the potential application of Nrf2 inducers in the prevention of pyroptosis as liver protective compounds.

## 1. Inflammasomes and Pyroptotic Cell Death

Inflammasomes are multi-protein signalling complexes that regulate the activation of inflammatory responses to microbial infection, and cellular damage, in addition to recognition of signals produced during altered homeostasis. An inflammasome is composed of an adaptor, an effector, and a sensor protein [1]. The assembly of inflammasomes is mediated by a sensor protein, which is usually a pattern recognition receptor (PRR) that oligomerizes following induction of activation by damage-associated molecular patterns (DAMPs), pathogen-associated molecular patterns (PAMPs), or homeostasis-altering molecular processes (HAMPs). Inflammasome activation results in the proteolytic cleavage and activation of pro-inflammatory Caspase-1, which triggers pro-inflammatory cytokines and induces pyroptotic cell death [2].

Based on their structural domains and their capacity to assemble inflammasomes, there are different receptor proteins that have been confirmed to be the sensor proteins for these complexes. The most studied are those formed by members of the nucleotide-binding oligomerization domain (NOD), leucine-rich repeat-containing receptor (NLR) family, comprising NLRP1, NLRP3, NLRP6, and NLRC4. These receptors have been documented as suitable for the formation of inflammasomes. Additionally, two non-NLR proteins, absent in melanoma 2 (AIM2) and Pyrin, also form functional inflammasomes [3]. NLRP3 is the most studied inflammasome as its dysregulation has been associated with the pathophysiology of the auto-inflammatory cryopyrin-associated periodic syndrome (CAPS), the development of chronic inflammatory conditions, neurodegenerative disorders, and metabolic diseases, including liver pathologies and fibrosis [4].

The first step of the canonical NLRP3 inflammasome activation involves the stimulation of different receptors, including Toll-like receptors (TLRs), and activation of the nuclear factor kappa B (NF-kB). The activation of these molecules, on the one hand, increases the transcription of *NLRP3*, *pro*-*IL1B*, and *pro*-*IL18*; while on the other hand, it induces post-translational modifications of NLRP3 in the form of (de-)ubiquitination or (de-)phosphorylation [5,6], necessary for its activation (Figure 1). A subsequent signal is required to induce activation of the NLRP3 inflammasome multiprotein complex. This second signal is usually mediated by DAMPs, such as crystalline particles (e.g., uric acid or cholesterol crystals) or detection of extracellular adenosine triphosphate (ATP) by the purinergic P2X7 receptor [7]. This signal is related to lysosomal destabilization or mitochondrial dysfunction, which induces the formation of reactive oxygen species (ROS), cellular metabolic changes, potassium (K^+^) efflux, and calcium (Ca^2+^) influx, that are upstream mechanisms for NLRP3 activation in response to most triggers [8] (Figure 1). The downstream signalling induced by these DAMPs in primed macrophages involves the activation of NLRP3 oligomers that recruit the adaptor inflammasome protein called apoptosis-associated speck-like protein containing caspase activation and recruitment domain (ASC) (Figure 1). This adaptor protein consists of two death-fold domains: a pyrin domain (PYD) and caspase activation and recruitment domain (CARD). ASC is recruited to the NLRP3 activated oligomer via PYD-PYD homotypic domain interaction. Further PYD-PYD interactions result in the formation of ASC filaments. The effector zymogen pro-Caspase-1 is then recruited to these ASC filaments by CARD-CARD interaction, and the close proximity of pro-Caspase-1 zymogens thereby induces an auto-proteolytic activation of Caspase-1 within the inflammasome complex, initiating the formation of the catalytically active protease Caspase-1 (Figure 1). Caspase-1 is the effector protein of the inflammasome complex that cleaves the preforms of the pro-inflammatory cytokines interleukin (IL)-1β and IL-18 into their mature and bioactive forms [9]. In addition, Caspase-1 cleaves Gasdermin D (GSDMD) and releases its necrotic and cytotoxic N-terminal domain from its repressor C-terminal domain [10,11] (Figure 1). Once GSDMD auto-inhibition is disrupted, the N-terminal domain of GSDMD binds to the plasma membrane and forms pores following homo-oligomerisation [12] (Figure 1). Both mature IL-1β and IL-18 are released through the GSDMD pore, as this pore has a negative conduit, and both cytokines present a basic surface after processing by Caspase-1 [13]. Upon permeabilization of the plasma membrane by GSDMD pores, cells undergo pyroptosis, a lytic process mediated by the oligomerization of the nerve injury-induced protein 1 [14]. During pyroptosis, other intracellular components, in addition to IL-1β and IL-18, are also released, including the alarmin high mobility group box 1 (HMGB1), mitochondrial DNA, or inflammasome oligomers [15,16], thereby resulting in a highly pro-inflammatory environment [17] (Figure 1). Various studies have associated excessive pyroptosis with multiple diseases, such as cardiovascular or immune diseases, disseminated intravascular coagulation in septic patients, and liver disorders [18,19].

The NLRP3 inflammasome is mainly expressed in myeloid cells, such as macrophages. However, it has also been described in other types of cells, such as endothelial cells, hepatocytes, or even hepatic stellate cells (HSCs); and its activation inducing pyroptosis has been associated with inflammation, fibrosis, and cell death in the liver [20].

## 2. Inflammasome and Pyroptosis as an Essential Inflammatory Pathway in Liver Diseases

Liver diseases are important contributors to global morbidity and mortality and are implicated in more than 45% of deaths in developed countries [21]. Diseases of the liver include conditions that cause any disturbance of its function resulting in illness. Sterile inflammation (in the absence of any pathogenic microorganisms) is a hallmark of the major diseases affecting the liver, including liver fibrosis, which results from chronic liver damage. The main causes of liver fibrosis in industrialized countries include alcohol abuse, non-alcoholic fatty liver disease (NAFLD), including non-alcoholic steatohepatitis (NASH), and also chronic viral infection. Liver fibrosis can lead to end-stage cirrhosis, where the liver parenchyma is substituted by scar tissue, wherein liver transplantation is the only option to recover normal liver functions [22]. In the presence of liver injury, some pro-inflammatory cytokines such as IL-1β are released, mainly by macrophages, activating HSCs through IL-1 receptor, turning into myofibroblasts and releasing a large amount of extracellular matrix, inducing the formation of scar tissue and leading to liver fibrosis [23]. Therefore, the NLRP3 inflammasome is considered to be a critical pathway for pro-inflammatory cytokine release in the liver and is strongly involved in the pathogenesis of liver fibrogenesis [24]. Constitutive activation of NLRP3 in an Nlrp3^A350v^ mutant mouse down-regulates metabolic pathways in hepatocytes and shifts HSCs toward a profibrotic state with an expression of collagen and extracellular matrix regulatory genes [25].

NLRP3 inflammasome is present in the cytoplasm of HSCs and is activated by various agents such as PAMPs (*S. mansoni*, *E. coli*, *S. japonicum*) [26,27,28] or through receptors such as P2X7R [29], angiotensin II receptors [30] or growth factor receptors [31]. In addition, NLRP3 activation in primary human hepatocytes leads to pyroptotic cell death and the release of inflammasome oligomeric particles into the extracellular space, resulting in the activation of HSCs by particle internalization [32]. In contrast, treatment with IL-10 and Jinlida, a traditional Chinese medicine, decreases the protein levels of NLRP3, IL1β, IL-18, and caspase-1, thus inhibiting the hepatocyte pyroptosis that ameliorated liver dysfunction [33,34]. In this regard, the regulation of pyroptosis by stress-inducible proteins such as Sestrin2, and transcription factors such as Ikaros together with SIRT1, has been reported to protect against liver injury through the downregulation of the inflammasome proteins [35,36]. More specifically, a recent study points to how treatment with phenethyl isothiocyanate, present as a natural compound in cruciferous vegetables and related to anti-cancer properties [37], can alleviate liver injury by reducing hepatocyte pyroptosis via direct inhibition of the cysteine 191 (Cys191) of GSDMD, which is necessary for the formation of GSDMD pores [38,39]. All of these results highlight the key role of pyroptosis in liver disease after NLRP3 inflammasome activation.

Numerous studies have demonstrated a link between the NLRP3 inflammasome and NAFLD [40], the condition in which fat builds up in the liver. NLRP3 inflammasome activation was indispensable in the fibrotic response in a murine model of NASH. In this regard, a methionine- and choline-deficient diet or high-fat diet evoked the overexpression of the inflammasome components NLRP3, ASC, caspase-1, and IL-1β [41]. Moreover, *Nlrp3* knockout mice fed to induce NASH showed a clear reduction in myeloid cell infiltration with the consequent decrease in liver inflammation and fibrosis [41]. Pyroptosis is involved in NASH development; that is, *Gsdmd* deficiency alleviates steatosis and inflammation, whereas *Gsdmd* overexpression promotes liver fibrosis [42].

Alcoholic liver disease (ALD), associated with long-term alcohol abuse, evolves from steatosis to severe stages such as alcoholic steatohepatitis (ASH) and alcoholic cirrhosis [43]. Similar to NAFLD, high levels of NLRP3 inflammasome and caspase-1 activation, and even serum IL-1β concentration, were found in ethanol-fed murine models [44]. The development of alcoholic steatosis has been associated with Kupffer cells (KCs), which present a significantly higher expression of inflammasome components compared to hepatocytes, and, in this regard, the silencing of the *Casp1* gene in KCs resulted in protection against steatosis, similar to that found in global *Casp1* knock-out mice [45,46]. Likewise, hepatocytes injured by chronic ethanol exposure induce the release of ATP and uric acid, which are classical DAMPs able to activate the NLRP3 inflammasome in KCs through the induction of mitochondrial ROS [47]. In this regard, alcoholic steatosis development has been prevented in *P2x7r* knock-out mice, in whom ATP signalling is blocked, or when uric acid is degraded by uricase treatment [45]. A mechanism of NLRP3 regulation of thioredoxin in response to ethanol has been discovered in ALD from human and mouse models. Both cases showed *Nlrp3*, *Pycard* (the ASC coding gene), and *Il1b* upregulation followed by high serum IL-1β levels compared to control animals [48]. These results are supported by the increased mRNA levels of NLRP3 inflammasome components, pro-inflammatory cytokines, and profibrotic genes in patients with alcohol-induced liver damage [49]. Moreover, prolonged alcohol intake is also associated with an altered gut microbiome, which is linked with intestinal dysbiosis, which in turn can lead to the translocation of bacterial products and bacteria from the gut to the liver, thus evoking a favourable environment for the activation of NLRP3 inflammasome [50,51]. Likewise, inhibition of poly (ADP-ribose) polymerase-1, a key mediator of liver inflammation and fibrosis [52], which protects against both NASH and ASH [53], can be directly related to the modulation of NLRP3 inflammasome and pyroptosis activation [54]. Furthermore, GSDMD plays a significant role in the evolution of liver steatosis into ASH. An increase in GSDMD was observed in the livers of animals with ASH compared to steatotic or control animals [55].

The hepatitis B (HBV) and C (HCV) viruses are associated with cirrhosis and hepatocellular carcinoma, where liver inflammation is observed in different stages of the disease [56]. In HBV, where the hepatitis X protein has a key role in liver inflammation by inducing the release of ROS, the NLRP3 inflammasome is activated and thus increases the expression of *IL1B*, *CASP1*, and *NLRP3* genes in the liver [57], followed by the secretion of ASC, IL-1β, IL-18, and HMGB1 in an environment of oxidative stress in hepatocytes [58]. This result is supported by high levels of IL-1β and IL-18 in plasma from patients with cirrhosis resulting from HBV [59]. Moreover, it has been shown that the HBV e antigen helps the virus to escape the host immune response by inhibiting NLRP3 inflammasome activation [60], while the hepatitis B core antigen seems to have a pro-inflammatory role by upregulating the expression and activation of the NLRP3 inflammasome, suggesting that the inflammasome is primed and activated in the first phases of infection [61]. Similarly, the HCV core protein has been shown to elicit NLRP3 inflammasome activation after the release of TNF-α with the consequent release of IL-1β by KCs to boost the inflammatory response of the liver [62,63].

In summary, inflammation is a major component in the progression of liver diseases, with important involvement of the NLRP3 inflammasome and pyroptosis. Therefore, the therapeutic targeting of NLRP3 or the proteins that accompany the formation of the NLRP3 inflammasome (such as ASC or Caspase-1) or even the released pro-inflammatory cytokines and the pore-forming protein GSDMD has garnered increasing consideration from the experts in the field and is an expanding area of research.

## 3. The Association of Oxidative Stress with NLRP3 Inflammasome Activation and Liver Diseases

Oxidative stress is characterised by the production of reactive oxygen and nitrogen species (ROS and RNS, respectively), including free radicals, and is augmented upon dysregulation of the production or removal mechanisms of the reactive species. Oxidative stress has been shown to be increased in the liver due to ethanol abuse, viral infection, and high-fat diets. In addition, the increase in the levels of ROS/RNS affects the initiation and progression of different hepatic pathologies [58,64]. As elaborated already, NF-κB activation is the first step in the canonical NLRP3 inflammasome activation (Figure 1), and ROS critically regulate the activity of the transcription factor NF-κB; for instance, the phosphorylation of Ser-276 of NF-κB p65 subunit is necessary for the positive transcription elongation factor b and is required for the expression of a subset of NF-κB-dependent genes [65]. However, there are other described mechanisms by which NF-κB can also be downregulated for ROS action. A specific cysteine residue in the p50 subunit of NF-κB is sensitive to oxidation by ROS. In particular, the oxidation of Cys-62 in the N-terminal DNA-binding domain of p50 inside the so-called Rel Homology domain [66] inhibits NF-κB DNA binding [67] and reduces NF-κB activation. Mechanistically, antioxidant treatment inhibits Ser-276 phosphorylation by regulating the ROS-dependent cAMP-dependent protein kinase catalytic subunit (PKAc) pathway, which contributes to NF-κB DNA-binding activity [68]. Accordingly, there exists reciprocal crosstalk between ROS production and NF-κB in the regulation of the inflammatory response (Figure 2), which affects the priming of the NLRP3 inflammasome.

Moreover, the role of ROS in the second step of canonical NLRP3 inflammasome activation has been extensively studied. In this regard, thioredoxin (TRX)-interacting protein (TXNIP) has been identified as a binding protein of NLRP3 that facilitates its activation due to ROS production [69] (Figure 2). Most canonical NLRP3 activators (monosodium urate crystals, asbestos, alum, silica, imiquimod, or extracellular ATP) increase the production of ROS through nicotinamide adenine dinucleotide phosphate (NADPH). However, the NLRP3 inflammasome activation remains unaffected even after the genetic knockdown of some of the components of the NADPH oxidase complex (NOX1, NOX2, and NOX4), suggesting that different ROS sources could be activated during NLRP3 activation [70,71,72]. In that regard, mitochondria are the main cellular generators of ROS, and the NLRP3 inflammasome activation is dependent on ROS accumulation upon inhibition of autophagy and mitophagy (mechanisms by which the ROS production is controlled) [73]. AN increase in ROS results in the oxidation of TRX and its release from TXNIP, leaving TXNIP free for binding to the NLRP3, inducing its activation [69]. However, although ROS are important in the activation of NLRP3, it must also be accompanied by the efflux of intracellular K^+^ [69].

In addition to controlling NLRP3 activation by ROS, GSDMD possesses a thiol group placed in the cysteine residues that can be modified by oxidative stress to form disulphide bonds (Cys-SS-Cys) among GSDMD subunits and stabilise pore formation in the plasma membrane. In fact, it has been proved that Cys191, placed in the active GSDMD^NT^ portion, needs to be essentially oxidised for the generation of GSDMD pores [38] (Figure 2).

## 4. Role of Antioxidant Compounds with Nrf2 Activation Properties in the Prevention of Liver Diseases

Nuclear factor erythroid 2-related factor 2 (Nrf2) is a transcription factor that activates the expression of numerous proteins with antioxidant properties. Nrf2 is usually associated with its cytoplasmic inhibitor Kelch-like ECH-associated protein 1 (Keap-1). In oxidative stress conditions, Keap-1 releases Nrf2, thus allowing its translocation into the nucleus. In the nucleus, Nrf2 binding to the antioxidant response element (ARE) sequence in the promoter regions induces the expression of several antioxidant proteins. Thus, Nrf2 is a master regulator with proven antioxidant and anti-inflammatory protective properties that are being studied for experimental treatment of diseases associated with oxidative stress and inflammation (reviewed in [74,75]).

The important role of NLRP3 inflammasome and pyroptosis in the inflammatory response during liver diseases implicate inflammasomes as promising drug targets to develop novel therapeutic approaches for liver diseases. Furthermore, inflammation and oxidative stress are usually closely associated with liver pathophysiological processes, suggesting a dual regulation of each other [76]. Therefore, attenuation of ROS production by antioxidant treatments is another promising approach for preventing liver inflammation. It has been reported that treatment with antioxidants decreases ROS and consequently reduces NLRP3 inflammasome activation in a carbon tetrachloride-induced acute liver injury model [77,78], further supporting the relationship between ROS generation and NLRP3 activation in liver disease. In the liver, NLRP3 upregulates Keap-1, the Nrf2 cytoplasmic inhibitor, and may trigger fibrogenesis by decreasing the Nrf2 activation, which increases ROS levels and, therefore, pyroptosis, which in turn exacerbates fibrosis [76]. Nrf2 is a master transcription factor that increases the expression of several endogenous antioxidant genes [79], and its pharmacological activators have demonstrated promising protective effects in liver disease [80], which may be attributed to its capacity to attenuate the NLRP3 inflammasome and pyroptosis [81]. Notably, recent studies have described a significant correlation between Nrf2 activation and the severity of inflammation, but not with the severity of steatosis, in a cohort of patients with non-alcoholic fatty liver disease, which may be due to a compensatory mechanism induced by an increase in oxidative stress associated with inflammation. However, the same study also describes how *Nrf2* activation in chronic liver disease in mice ameliorates fibrogenesis, which is a key mechanism in the progression of hepatocellular carcinogenesis, thus demonstrating the hepatoprotective properties of Nrf2 activity [82].

In the last three years, a significant number of publications have reported that some bioactive compounds, such as polyphenols and triterpenoids, extracted from medicinal herbs or functional foods, have demonstrated important antioxidant and liver-protective properties [83,84,85], in addition to their ability to activate Nrf2 (Figure 2 and Table 1).

Curcumin is a polyphenol popularly known as a colourant and food additive in the European Union that functions as a free radical scavenger with remarkable protective and therapeutic effects on liver diseases associated with inflammatory and oxidative processes. These effects may be, at least in part, attributed to an increase in the activity of the Nrf2 pathway [86]. Naringenin is a flavonoid with antioxidant and anti-inflammatory properties, which when used in treatment has shown significant beneficial effects in liver diseases [87]. Moreover, other studies have demonstrated the capacity of Naringenin to induce Nrf2 pathways in astroglia, suggesting that this molecular mechanism may also be present in other organs such as the liver [88]. Mycelium polysaccharides extracted from *Pleurotus geesteranus* (IMPP), a commonly consumed mushroom, have received a great deal of attention for its antioxidant and anti-inflammatory effects with demonstrated beneficial effects, suggesting its role as a possible functional food in acute alcoholic liver diseases [89]. Mycelium polysaccharides from IMPP have also demonstrated abilities to activate the heme oxygenase 1 (HO-1)/Nrf2 pathway increasing antioxidant enzymes activity to protect the liver [90]. Silymarin is an extract that has also been used for over centuries to treat liver diseases and has proven effects in animal models [91]. Astaxanthin is a carotenoid that is a coloured fat-soluble pigment used as food colouring for seafood such as salmon or shrimps. It has also shown to be an Nrf2 inducer with strong antioxidant and anti-inflammatory properties, with proven preventive and therapeutic effects on liver fibrosis, liver tumours, liver ischemia-reperfusion injury, non-alcoholic fatty liver, and other related diseases [92]. Fucoidan is a sulfuric acid group-rich polysaccharide found in brown algae with the capacity to induce Nrf2 activation [93]. Furthermore, it has been considered a compound with a promising potential clinical application for the development of future follow-up medication for liver diseases due to its multiple biological properties such as antithrombotic, antiviral, antioxidant, and immune function promoting characteristics, in addition to its demonstrated protective effects on multiple liver diseases [94]. Other polyphenols present in plants, such as rosmarinic acid [95] and ellagic acid [97], also demonstrate effective hepatoprotective properties, which may also be due to their capacity to regulate Nrf2 pathways [96,98]. Melatonin is a pineal gland hormone which is also present in different plants. It is a known antioxidant and Nrf2 inducer [100], and previous reports have shown its protective effects not only in the early stages of fibrosis but also in end-stage liver cirrhosis [99]. *Annona squamosa* is another plant with enormous therapeutic potential for liver diseases. Its seed extract has shown protective effects against alcohol-induced liver injury in Sprague Dawley rats [101]. Indeed, it has been reported that bardoxolone methyl offers an effective pharmacological approach to increasing Nrf2 activity and mitigating cholestasis in hepatic ischemia-reperfusion injury [102]. Furthermore, additional herbal products such as caffeine and stevia have proven antioxidant properties, in part due to their capacity to induce Nrf2 activation [104,106]. Studies in rats demonstrated the hepatoprotective properties of low doses of caffeine on CCl4-induced liver damage [105], and stevia treatment prevents experimental cirrhosis, thus indicating that both products also be also useful in the clinical setting to prevent liver damage in chronic liver diseases [103,104].

Overall, even considering that many of these drugs are multitarget and exert their protection through different signaling pathways, the preclinical and clinical studies discussed so far highlight the fact that all antioxidant compounds quoted have the common characteristic of activating Nrf2. They further indicate that the Nrf2 pathway is one of the key pathways for designing effective targets to develop new treatments for protection against liver injury. However, aberrant activation of Nrf2 may also be associated with poor prognosis in other pathologies such as cancer. It has been described that Nrf2 constitutive activation induces pro-survival genes and promotes cancer cell proliferation by metabolic reprogramming in various cancers cells types, which could induce apoptosis repression, and increase the capacity of self-renewal of cancer stem cells (reviewed in [107]), thus suggesting that excessive or prolonged Nrf2 activation may contribute to tumorigenesis. Nevertheless, attention should also be paid to the possible side effects of these antioxidants, which are discussed in the following section.

## 5. Clinical Trials with Nrf2-Activating Antioxidants: Results and Side Effects

Oxidative stress is a topic that has motivated thousands of scientific publications and has retained the interest of the scientific community during the last 20 years as ROS is evidently involved in the pathogenesis of numerous diseases. Several antioxidant drugs have undergone clinical trials for the last two decades to determine if they can prevent the deleterious effects of oxidative stress in some diseases. However, in numerous studies, the protective effect of antioxidant treatment has not been validated. Moreover, numerous side effects have been described that are associated with these antioxidant treatments. For example, a recent meta-analysis study on the effects of N-acetylcysteine in acute liver failure was found to be inconclusive regarding the beneficial effects of N-acetylcysteine as compared with placebo [108]. Moreover, vitamin E treatment has been shown to increase mortality rate as well as important side effects [109]. The evaluation of a phase 3 clinical trial with bardoxolone methyl, an Nrf2 inducer, in diabetic patients with chronic kidney disease was terminated prematurely on the recommendation of the safety monitoring committee due to the strong adverse safety signals associated with the treatment, such as increased rates of heart failure, cardiovascular events, high blood pressure levels, heart rate, and albuminuria compared with the placebo group [110]. Moreover, early studies on bardoxolone methyl have shown that it increases serum alanine aminotransferase (ALT) and aspartate aminotransferase (AST) in patients with diabetic renal diseases, which are commonly considered to be reflective of liver injury [111].

## 6. Conclusions and Future Perspectives

The terms inflammasome and pyroptosis have only been recently adopted in the 21st century. However, despite the short time frame of these findings, more than ten thousand research articles have been published on them, with an exponential rise in the last five years. Moreover, the relationship between inflammasome activation and pyroptosis with a growing amount of diseases such as inflammatory bowel diseases, rheumatoid arthritis, Parkinson’s, Alzheimer’s, obesity, myocardial infarction, and even liver diseases has been established in the recent years. This makes the role of inflammasome and pyroptosis one of the most relevant topics of current times.

At present, the development of liver diseases cannot be prevented with existing pharmacological therapies. Therefore, new approaches to therapy and targets are urgently required. Concrete evidence suggests that attenuation of the inflammatory response evoked by NLRP3 inflammasome activation may be a critical step for the restoration of the proper balance between pro- and anti-fibrotic signalling pathways. Accordingly, it represents one of the main therapeutic targets in the prevention of chronic liver diseases. Activation of the inflammasome pathway and pyroptosis has been related to oxidative stress; for instance, ROS production is necessary for complete activation of NLRP3, not only for the priming and engagement of the canonical NLRP3 inflammasome but also for GSDMD pore formation in the plasma membrane. Therefore, when different antioxidants are used to decrease or reverse liver fibrosis, they could affect NLRP3 activation and the oxidation of the cysteine residues of GSDMD essential for complete functional pore formation, thereby disabling the function of NLRP3 in liver diseases. In this way, the application of Nrf2 activators as antioxidants in liver pathologies is currently trending fast and provides strong evidence suggesting that the Nrf2 pathway is a promising target for future novel treatment of liver diseases.

Numerous studies have demonstrated the controversial role of ROS in different cellular functions [112] and have suggested that total and chronic ROS suppression could have crucial consequences on physiological functions, indicating that the time, location, and amount of ROS production are critical in determining their deleterious or beneficial effects on cellular metabolism [112]. In this regard, our previous studies have shown that DJ-1 protein increases the Nrf2 concentration by limiting Nrf2 degradation [113] and that ND-13, a peptide based on the amino-acid sequence of DJ-1, has protective effects on inflammation and fibrosis [114]. These pieces of evidence suggest that increasing Nrf2 concentration without inducing its activation could strengthen the activity of Nrf2 in pathological conditions and maintain the correct cellular redox balance. Thus, this new approach may be useful in order to avoid the side effects associated with chronic Nrf2 activation described in previous clinical trials.

The current challenge is to design new approaches to reinforce Nrf2 actions while simultaneously maintaining the correct redox homeostasis in the tissues. Therefore, in this context, we propose that inducing Nrf2 expression or avoiding Nrf2 degradation without affecting Nrf2 regulatory mechanisms could be a new promising approach for strengthening Nrf2 activity where the oxidative stress is localized and for preventing the unwanted side effects of systemically chronic Nrf2 activation.

## Figures and Tables

**Figure 1 antioxidants-11-00870-f001:**
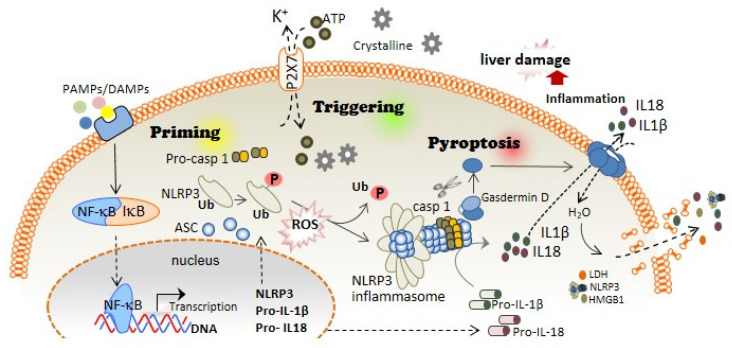
**NLRP3 inflammasome and pyroptosis activation induces liver damage**. The canonical activation of the NLRP3 inflammasome is differentiated in two steps: priming and triggering. Priming; PAMPs and DAMPs stimulate TLR and other receptors, which induce the NF-κB translocation to the cell nucleus, thus increasing the expression of the different inflammasome components: NLRP3, pro-IL-1β, and pro-IL-18 expression. During priming signaling, NLRP3 also undergoes different post-translational modifications that facilitate its activation, including phosphorylation/dephosphorylation and de-ubiquitination. Triggering; DAMPs such as extracellular ATP or crystalline structures induce the activation of the NLRP3 inflammasome oligomer, which leads to the activation of Caspase-1. Caspase-1 cleaves Gasdermin D in addition to pro-IL-1β and pro-IL-18, which turn into mature IL-1β and IL-18. Pyroptosis; The N-terminal fragment of Gasdermin D generated after Caspase-1 cleavage forms oligomeric pores in the plasma membrane allowing the release of pro-inflammatory cytokines such as IL-1β and IL-18 into the extracellular space. Additionally, Gasdermin D pores lead to water influx into the cell, cell swelling, and cell lysis mediated by the ninjury-1 protein, thus increasing the inflammatory response by releasing inflammatory products from the intracellular space.

**Figure 2 antioxidants-11-00870-f002:**
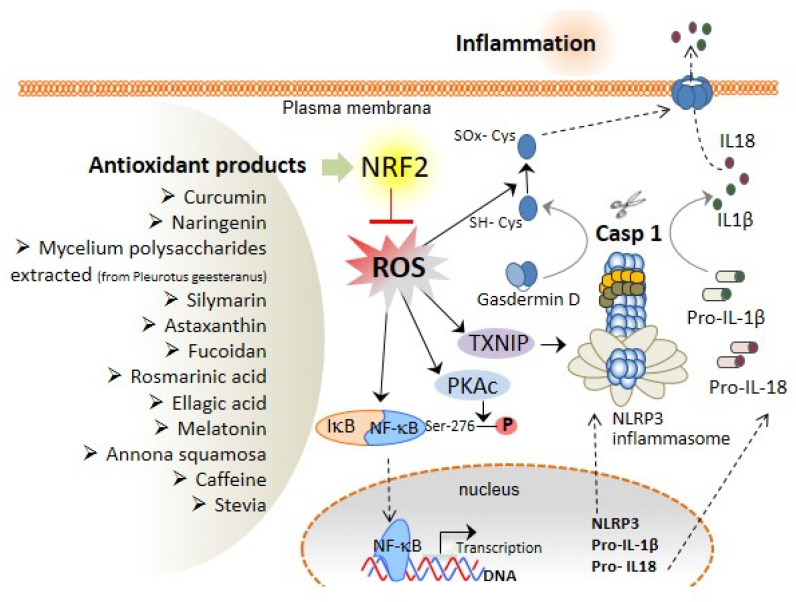
**Role of ROS in NLRP3 inflammasome activation and liver damage.** inflammasome activation and pyroptosis execution, thus inducing inflammation. Antioxidant products increase Nrf2 activity, which attenuates ROS production and prevents the oxidation of gasdermin D, TXNIP, and NF-kB. The oxidation of some amino acid residues in these compounds is needed for inflammasome activation and pyroptosis induction; consequently, antioxidant products may prevent the inflammatory response via Nrf2 activation.

**Table 1 antioxidants-11-00870-t001:** List of the antioxidant compounds which induce Nrf2 pathway and its effect on liver pathologies.

Drug	Source	Type of Molecule	Effect on Liver Pathologies	Biological System	References
Curcumin	Plants	Polyphenol	NASH; ALD; Liver fibrosis	Rat; Mice	[86]
Naringenin	Plants	Flavonoid	ALD; Liver fibrosis; Diabetes-induced hepatotoxicity; Liver Cancer	Rat; Mice; Human; Rabbit	[87,88]
Pleurotus geesteranus	Mushroom	Mycelium polysaccharide	ALD; Chronic liver injury	Mice	[89,90]
Silymarin	Plants	Extract	Diabetes Type 1-induced hepatotoxicity	Rat	[91]
Astaxanthin	Seafood, microalgaes, yeasts	Carotenoid	Liver fibrosis; NAFLD; Liver Cancer, Liver cirrhosis; Hepatic IRI	Rat; Mice; Human	[92]
Fucoidan	Brown algae	Sulfuric acid-group-rich polysaccharide	Acute liver injury; Viral Hepatitis; Liver fibrosis; Liver Cancer; Hepatic IRI; NAFLD	Rat; Mice; Human	[93,94]
Rosmarinic acid	Plants	Caffeic acid ester	Hepatic IRI	Rat; Mice	[95,96]
Ellagic acid	Plants	Polyphenol	Liver fibrosis; NAFLD; Viral Hepatitis	Rat; Mice; Human	[97,98]
Melatonin	Plants	Pineal gland hormone	Liver fibrosis; NAFLD; NASH; Liver cirrhosis; Liver injury	Rat; Mice	[99,100]
Annona squamosa	Plants	Seed extract	ALD	Rat	[101]
Bardoxolone	Semi-synthetic	Triterpenoid	Hepatic IRI	Rat	[102]
Stevia	Plants	Diterpene	Liver cirrhosis	Rat; Mice	[103,104]
Caffeine	Plants	Alkaloid (Xanthine)	Liver injury	Rat; Mice	[105,106]

ALD, Alcoholic liver disease; IRI, Ischemia-reperfusion injury; NAFLD, Non-alcoholic fatty liver disease; NASH, Non-alcoholic steatohepatitis.

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
