# Peer review of "NLRP3 Inflammasome and Pyroptosis in Liver Pathophysiology: The Emerging Relevance of Nrf2 Inducers"

_antioxidants, 2022, doi:10.3390/antiox11050870_

Round 1
Reviewer 1 Report
The authors performed all requested changes. The manuscript has been strongly improved.
Reviewer 2 Report
I have revised the revised version of the manuscript and the point-by-point response and found that the authors improved their manuscript according to the reviewer's suggestions.
My recommendation is acceptance of the manuscript in the present form.
This manuscript is a resubmission of an earlier submission. The following is a list of the peer review reports and author responses from that submission.
Round 1
Reviewer 1 Report
Several pathological liver conditions are associated with increased inflammatory reactions triggered by inflammasome activation and pyroptosis. In this review manuscript, the authors describe inflammasome activation and its role in liver diseases. They point out that reactive oxygen species are (ROS) play a major role in these processes and indicate that antioxidative mechanism triggered via the protein Nrf2 can counteract the generation of ROS and thereby suppress liver injury. Therefore, the authors suggest that Nrf2 inducers might constitute suitable therapeutic approaches to prvent inflammasome activation and liver diseases.
This is an interesting report which provides a nice overview on mechanisms of inflammasome activation and its role in liver diseases.
Several points need to be respected.
- Line 59: Since IL-1b and IL18 are generated from their pre-stages during inflammasome activation, it might be better to use pre-IL1B and pre-IL18 at this point. This is also more consistent with the figure legend.
- Line 60: The authors indicate “post-transcriptional” modifications of NLRP3 which is not consistent with the figure where “post-translational” modifications are indicated. I assume that the authors meant phosphorylation and ubiquitination and would therefore suggest to use “post-translational” at both occasions. In addition, I would suggest to provide and explain the distinct modifications in brief.
- Some abbreviations are not explained in the text (e.g. KC) and no abbreviation index is provided. Please take care for that.
- Line 137: The authors describe that IL10 ameliorates liver dysfunction. This statement requires a little more information. What is the source of IL-10 and how can it inhibit pyroptosis.
- Line 209: It is somehow confusing to start with an example where ROS inhibit NF-kB activation. In context with liver diseases one would expect that mainly NF-kB activation is important which is explained in the subsequent paragraph. I would suggest to exchange the arrangement and indicate that ROS can also inhibit NFKB after showing that it activates NFkB.
- Line 214. The PKA pathway mentioned here might also be shown in the figure.
- Line 262: The authors point out that NLRP3 inhibits NRF2. This is not clear and not sufficiently explained, also not shown in the figure.
- Some more properties of Nrf2 structure and function should be provided.
- Section 4, The role of antioxidant compounds with Nrf2 activation properties is a kind of listing of different substances. There is less information about potential mechanism of the different drugs and the biological systems in which they were used.
- In general, the authors suggest Nrf2 inducers as potential measure to indirectly inhibit NLRP3 and pyroptosis but do not discuss other possibilities to inhibit them. This should be implemented into the review.
- Section 5: The authors indicate “important side effects” or strong adverse safety signals” associate with the Nrf2 activators but do not state which side effects are meant. This information should be added.
- Several corrections of English grammar are necessary.
Reviewer 2 Report
Major comments:
- PARP is a major player in liver pathophysiology and pyroptosis. Please elaborate with updated references such as J Hepatol 2017 Mar;66(3):589-600; Cells. 2020 Oct; 9(10): 2286; Hepatology 2014 May;59(5):1998-2009.
- NRF2 is a double edge sword in oxidative stress biology , authors should elaborate that aspect.
- 3. Many recent important references on pyroptosis are also missing
Reviewer 3 Report
check references 34, 40, 73, 78, 80,
Why such a sustantial number of reviews or comments in this manuscript?
Can the authors provide a table with the antioxidants described in the manuscript?
Hepatic stellate cells (HSCs) are crucial in liver fibrosis, is there any role for NLRP3 and/or GSDMD in HSCs?
Can the authors comment on free fatty acid induced cell death such as the lipotoxic palmitic acid as this would fit in the NAFLD/NASH part?
Reviewer 4 Report
Title: NLRP3 inflammasome and pyroptosis in liver pathophysiology: the emerging relevance of Nrf2 inducers
This is a very interesting review dealing with the importance of NLRP3 inflammasome and pyroptosis on liver diseases. The authors describe the general mechanisms of NLRP3 activation and the role of this factor in liver pathophysiology. Additionally, the Nrf2 signaling pathway is reviewed in the context of hepatic disorders and the crosstalk of NRLP3 and Nrf2 signaling pathways. Several plant-derived drugs that upregulate the Nrf2 pathway are described. This is an original review dealing with a hot topic. However, I have some concerns:
- The authors describe the hepatoprotective activity of several phytodrugs reducing their propertied on their ability to upregulate the Nrf2 signaling pathway. Many of these drugs are multitarget and exert their activity on several signaling pathways that deserve attention. For example, the curcumin ability to directly block NF-kappaB activation is well-recognized, etc.
- A table with the list of drugs that induce the Nrf2 pathway with the effect on the liver and reference will be useful. Include other herbal drugs such as caffeine, stevia (stevioside and rebaudioside).
- Perhaps authors may improve their review by including profibrogenic pathways and factors, such as TGF-beta, Smads, CTGF, PDGF in the context of oxidative stress and the effects of antioxidants and Nrf2 inducers.
- English edition is needed.